# Deep Learning and Gastric Cancer: Systematic Review of AI-Assisted Endoscopy

**DOI:** 10.3390/diagnostics13243613

**Published:** 2023-12-06

**Authors:** Eyal Klang, Ali Soroush, Girish N. Nadkarni, Kassem Sharif, Adi Lahat

**Affiliations:** 1Division of Data-Driven and Digital Medicine (D3M), Icahn School of Medicine at Mount Sinai, New York, NY 10029, USA; eyal.klang@sheba.health.gov.il (E.K.); ali.soroush@mountsini.org (A.S.); girish.nadkarni@mountsinai.org (G.N.N.); 2The Charles Bronfman Institute of Personalized Medicine, Icahn School of Medicine at Mount Sinai, New York, NY 10029, USA; 3ARC Innovation Center, Sheba Medical Center, Affiliated with Tel Aviv University Medical School, Tel Hashomer, Ramat Gan 52621, Tel Aviv, Israel; 4Department of Gastroenterology, Sheba Medical Center, Affiliated with Tel Aviv University Medical School, Tel Hashomer, Ramat Gan 52621, Tel Aviv, Israel; kassemsharif@gmail.com

**Keywords:** gastric cancer, deep learning, artificial intelligence, systematic review, endoscopy

## Abstract

Background: Gastric cancer (GC), a significant health burden worldwide, is typically diagnosed in the advanced stages due to its non-specific symptoms and complex morphological features. Deep learning (DL) has shown potential for improving and standardizing early GC detection. This systematic review aims to evaluate the current status of DL in pre-malignant, early-stage, and gastric neoplasia analysis. Methods: A comprehensive literature search was conducted in PubMed/MEDLINE for original studies implementing DL algorithms for gastric neoplasia detection using endoscopic images. We adhered to the Preferred Reporting Items for Systematic Reviews and Meta-Analyses (PRISMA) guidelines. The focus was on studies providing quantitative diagnostic performance measures and those comparing AI performance with human endoscopists. Results: Our review encompasses 42 studies that utilize a variety of DL techniques. The findings demonstrate the utility of DL in GC classification, detection, tumor invasion depth assessment, cancer margin delineation, lesion segmentation, and detection of early-stage and pre-malignant lesions. Notably, DL models frequently matched or outperformed human endoscopists in diagnostic accuracy. However, heterogeneity in DL algorithms, imaging techniques, and study designs precluded a definitive conclusion about the best algorithmic approach. Conclusions: The promise of artificial intelligence in improving and standardizing gastric neoplasia detection, diagnosis, and segmentation is significant. This review is limited by predominantly single-center studies and undisclosed datasets used in AI training, impacting generalizability and demographic representation. Further, retrospective algorithm training may not reflect actual clinical performance, and a lack of model details hinders replication efforts. More research is needed to substantiate these findings, including larger-scale multi-center studies, prospective clinical trials, and comprehensive technical reporting of DL algorithms and datasets, particularly regarding the heterogeneity in DL algorithms and study designs.

## 1. Introduction

Gastric cancer (GC) is a significant health issue worldwide [1]. The global prevalence of gastric cancer is increasing significantly, emphasizing the need for improved detection methods. Currently, gastric cancer ranks as the fifth most common malignant cancer and the fourth leading cause of cancer-related mortality worldwide. Despite a decline in incidence rates, the global burden of this malignancy is projected to increase by 62% by 2040 [2]. This escalation is further highlighted by predictions from the International Agency for Research on Cancer (IARC), which forecasts an increase to about 1.8 million new cases and approximately 1.3 million deaths by 2040, representing increases of about 63% and 66%, respectively, compared with 2020 [3].

The five-year survival rate for stage Ia cancer is over 95%, compared to 66.5% and 46.9% for stages II and III, respectively [4]. These survival differences underscore the importance of detecting treatable early-stage neoplastic lesions in the stomach. However, subtle surface changes suggestive of gastric neoplasias can be very challenging to identify endoscopically [5]. An estimated 5–10% of individuals diagnosed with gastric cancer had a negative upper endoscopy within 3 years, suggesting a failure to diagnose early-stage gastric neoplasias [5,6,7]. Procedures performed by non-expert endoscopists have an additional 14% decrease in the absolute gastric neoplasia detection rate compared to those performed by expert endoscopists [6]. Recent advancements in artificial intelligence (AI)-assisted endoscopy offer hope. Computer-aided detection (CADe) and diagnosis (CADx) AI systems aim to improve the real-time detection of gastric neoplasias by detecting subtle neoplastic changes in gastric mucosa with high accuracy [8,9].

Deep learning is a sophisticated branch of artificial intelligence that uses algorithms inspired by the human brain, known as artificial neural networks. These networks consist of layers of nodes that transform input data, enabling the machine to learn from large datasets. The learning process involves adjusting the network’s internal parameters to reduce prediction errors, utilizing methods like backpropagation and gradient descent. Deep learning is distinguished by its use of numerous layers that enable the recognition of complex patterns, making it ideal for applications in image and speech recognition, among others [10].

The proliferation of AI tools can be largely attributed to advances in computational power, algorithmic improvements, and the abundance of data available in the digital age. These tools have found applications across various sectors, including healthcare, automotive, finance, and customer service. The development of accessible frameworks like TensorFlow and PyTorch has democratized AI, allowing a broader range of users to develop AI models. Additionally, the integration of AI into products and services by many companies has fueled further demand for these tools.

Convolutional neural networks (CNNs) are a type of deep learning (DL) model used in computer vision. CNNs are trained using vast quantities of annotated images and videos that are labelled to facilitate particular tasks such as image classification or segmentation. Via the segmentation process, which begins with the collecting and preprocessing of gastric images, followed by expert annotation to mark cancerous tissues, the AI model is trained on these annotated images to learn the distinguishing features of neoplastic tissue. Once trained, the model can segment neoplastic areas in new images, aiding in diagnosis and treatment planning [11].

Once trained, these models can independently analyze new visual data, effectively detecting (CADe) or diagnosing (CADx) gastric neoplasias [12]. The real-world use of CNN-assisted endoscopy has shown encouraging results for identifying upper gastrointestinal neoplasms, including GC [7]. AI-based systems can also estimate GC’s invasion depth, which is crucial for treatment planning [13,14].

In many cases, endoscopic AI systems match or surpass the diagnostic performance of experienced endoscopists [15,16,17,18].

Real-time segmentation models have also successfully distinguished between gastric intestinal metaplasia (GIM) and healthy stomach tissue [17]. AI technology has the potential to revolutionize gastrointestinal endoscopy, improving patient outcomes. Yet, integrating CADe/CADx systems into clinical workflows is not without challenges. Algorithm training bias, the management of human–AI interaction complexities, and dealing with false-positive outcomes all need to be addressed to ensure successful AI implementation [9].

This review aims to probe the current literature on DL’s role in gastric neoplasia detection and diagnosis. We aim to analyze the strengths and weaknesses of CNN-based systems used for endoscopic evaluation of gastric neoplasias, and to discuss the challenges of integrating these systems into clinical practice.

## 2. Methods

We carried out this systematic review following the Preferred Reporting Items for Systematic Reviews and Meta-Analyses (PRISMA) guidelines.

### 2.1. Search Strategy

We conducted a comprehensive search of PubMed/MEDLINE from their inception until May 2023. We used a combination of Medical Subject Headings (MeSH) and free-text terms related to “gastric cancer”, “gastric neoplasm”, “stomach neoplasm”, “stomach cancer”, “gastric carcinoma”, “artificial intelligence”, “deep learning”, “machine learning”, and “endoscopy”. We restricted the search to articles published in English. We manually checked the reference lists of eligible articles for any additional relevant studies.

### 2.2. Selection Criteria

We included studies that met the following criteria: 1—original research articles, 2—application of DL algorithms for gastric neoplasia detection or diagnosis, 3—use of endoscopic images for analysis, 4—provided quantitative diagnostic performance measures, and 5—compared AI performance with that of human endoscopists where possible.

We excluded studies if they: 1—were reviews, case reports, editorials, letters, or conference abstracts, 2—did not independently assess gastric neoplasias, 3—did not use endoscopic imaging, or 4—lacked sufficient performance measures.

### 2.3. Data Extraction

Two reviewers independently extracted data using a pre-defined extraction form using Microsoft Excel. The data included variables like the first author’s name, publication year, and study design, among others. Disagreements were resolved through discussion or consultation with a third reviewer.

### 2.4. Data Synthesis and Analysis

We performed a narrative synthesis to summarize the studies’ findings. Because of the expected heterogeneity in AI algorithms, imaging techniques, and study designs, we did not plan a meta-analysis. We presented the results in a tabular format and discussed the key findings.

## 3. Results

We identified 42 studies that met the inclusion criteria. The designs, sample sizes, and algorithms used varied across the studies. Table 1 summarizes all studies included in the review and offers data on the research subject, clinical task, study design, sample size, detection/classification, objective, AI model/algorithm, AI model performance, clinical implication, and subject. The flowchart delineating the selection procedure of the studies included is depicted in Figure 1.

All sensitivity, specificity, and accuracy values represent the AI models’ performance.

### 3.1. Gastric Neoplasm Classification

Several studies utilized DL to classify gastric endoscopic images. Lee JH et al. [19] used Inception, ResNet, and VGGNet to differentiate between normal, benign ulcer, and cancer images, achieving high areas under the curves (AUCs) of 0.95, 0.97, and 0.85, respectively.

Notably, Inception, ResNet, and VGGNet are influential architectures in the field of deep learning, particularly for image processing tasks:

Inception: Known for its efficiency and depth, the Inception architecture, especially its popular version Inception v3, uses a combination of different-sized convolutional filters within the same layer (called a “network in network” approach). This allows it to capture spatial hierarchies at different scales and reduces computational cost.

ResNet (Residual Network): This architecture introduces residual learning to alleviate the vanishing gradient problem in very deep networks. By using skip connections that bypass one or more layers, ResNet can effectively train networks with many layers, including popular variants like ResNet50 and esNet101.

VGGNet (Visual Geometry Group Network): VGGNet, particularly VGG16 and VGG19, is known for its deep but simple architecture, using multiple convolutional layers with small-sized filters (3 × 3) before pooling layers. Its uniform architecture makes it easy to understand and implement, and it has been highly influential in demonstrating the effectiveness of depth in convolutional neural networks [19].

Cho BJ et al. [20] used Inception-Resnet-v2 for gastric neoplasm classification, reaching 84.6% accuracy. Three studies focused on diagnosing GC.

The ResNet50 model is a variant of the ResNet architecture, which is widely used in deep learning for image recognition and processing tasks. It is 50 layers deep, hence the ‘50’ in its name. One of the key features of ResNet is the use of “residual blocks” that help in combatting the vanishing gradient problem in deep neural networks, enabling the training of much deeper networks.

In a ResNet50 model, these residual blocks consist of layers with skip connections that allow the activation from one layer to be fast-forwarded to a later layer, bypassing one or more layers in between. This design helps in preserving the learning and signal from the initial layers to the deeper layers in the network. ResNet50, specifically, is known for its balance between depth and complexity, making it a popular choice for many image recognition tasks [21].

ResNet50 was applied to Magnifying Endoscopy with Narrow-Band Imaging (ME-NBI) images [21], resulting in 98.7% accuracy. Yusuke Horiuchi et al. [22] achieved 85.3% accuracy in differentiating early gastric cancer (EGC) from gastritis using a 22-layer CNN. Another study [23] achieved 92% sensitivity and 75% specificity in distinguishing malignant from benign gastric ulcers using a CNN.

Liu Y et al. [24] used the CNN Xception model enhanced by a residual attention mechanism to automatically classify benign and malignant gastric ulcer lesions in digestive endoscopy images.

The Xception model is an advanced convolutional neural network (CNN) architecture that builds upon the principles of Inception networks. The key innovation in Xception is the concept of depthwise separable convolutions. This approach separates the convolution process into two parts: a depthwise convolution that applies a single filter per input channel, followed by a pointwise convolution that applies a 1 × 1 convolution to combine the outputs of the depthwise convolution.

This architecture allows Xception to have fewer parameters and computations than a traditional convolutional approach, while maintaining or improving model performance. Xception is particularly effective for tasks in computer vision, demonstrating strong performance in image classification and other related tasks.

The enhanced Xception model achieved an accuracy of 81.4% and F1 score of 81.8% in diagnosing benign and malignant gastric ulcers.

### 3.2. Gastric Neoplasm Detection

Two studies [25,26] used SSD-GPNet and YOLOv4 for detecting gastric polyps, both achieving high mean average precision (mAP). YOLO, which stands for “You Only Look Once”, is a revolutionary AI algorithm used for real-time object detection in computer vision. It streamlines the traditional two-step object detection process into a single step utilizing a single convolutional neural network. This approach allows YOLO to quickly identify and classify multiple objects within an image by dividing it into a grid and predicting bounding boxes and class probabilities for each grid cell. Renowned for its speed, YOLO also maintains a commendable level of accuracy, although it is slightly less effective with small objects. Its applications range from autonomous vehicles to real-time surveillance. Over time, various iterations of YOLO have emerged, each enhancing its speed, accuracy, and detection capabilities, making it a crucial tool in the field of AI-assisted object detection.

The SSD-GPNet was shown to significantly increase polyp detection recalls by more than 10% (*p* < 0.001), particularly in the detection of small polyps [25], and YOLOv4 was shown to have an 88.0% mean average precision [26].

With respect to gastric neoplasm detection, Lianlian Wu et al. [27] showed a decreased miss rate using AI: the incidence of missed gastric neoplasms was significantly reduced in the AI-first group compared to the routine-first group (6.1%, [3 out of 49 patients] versus 27.3%, [12 out of 44 patients]; *p* = 0.015). Hongliu Du et al. [16] achieved the highest accuracy using a multimodal (white light and weak magnification) model, achieving 93.55% accuracy in prospective validation and outperforming endoscopists in the evaluation of multimodal data (90.0% vs. 76.2%, *p* = 0.002). Furthermore, when assisted by the multimodal ENDOANGEL-MM model, non-experts experienced a significant improvement in accuracy (85.6% versus 70.8%, *p* = 0.020), reaching a level not significantly different from that of experts (85.6% versus 89.0%, *p* = 0.159).

Xu M et al. [28] retrospectively analyzed endoscopic images from five Chinese hospitals to evaluate the efficacy of a computer-aided detection (CADe) system in diagnosing precancerous gastric conditions. The CADe system demonstrated high accuracy, ranging between 86.4% and 90.8%. This was on par with expert endoscopists and notably better than non-experts. A single-shot multibox detector (SSD) was used by one study [29] for gastric cancer detection, achieving 92.2% sensitivity. Another study [30] achieved 58.4% sensitivity and 87.3% specificity in early detection using a CNN.

Leheng Liu et al. [11] developed a computer-aided diagnosis (CAD) system to assist in diagnosing and segmenting Gastric Neoplastic Lesions (GNLs). The study utilized two CNNs: CNN1 for diagnosing GNLs and CNN2 for segmenting them. CNN1 demonstrated excellent diagnostic performance with an accuracy of 90.8%. It also achieved an impressive area under the curve (AUC) of 0.928. The use of CNN1 improved the accuracy rates for all participating endoscopists compared to their individual diagnostic methods. CNN2, focused on segmentation, also performed well, achieving an average intersection over union (IOU) of 0.584 and high values for precision, recall, and the Dice coefficient.

### 3.3. Assessment of Tumor Invasion Depth

Bang CS et al. [31] utilized AutoDL for classifying the invasion depth of gastric neoplasms with 89.3% accuracy.

VGG-16 is a convolutional neural network model proposed by the Visual Graphics Group (VGG) from the University of Oxford. It is composed of 16 layers (hence the name VGG-16), including 13 convolutional layers and 3 fully connected layers. One of the key characteristics of VGG-16 is its use of a large number of convolutional layers with small-sized filters (3 × 3), which allows it to capture complex patterns in the data while keeping the computational complexity manageable [32].

Hong Jin Yoon et al. [32] used VGG-16 for assessing tumor invasion depth in EGC, achieving highly discriminative AUC values of 0.981 and 0.851 for EGC detection and depth prediction, respectively. Of the potential factors influencing AI performance when predicting tumor depth, only histologic differentiation showed a significant association. Undifferentiated-type histology was associated with lower-accuracy AI predictions.

Kenta Hamada et al. [33] used ResNet152 for and achieved 78.9% accuracy in diagnosing mucosal cancers. Yan Zhu et al. [13] achieved an AUC of 0.94 using ResNet50. When using a threshold value of 0.5, the system demonstrated a sensitivity of 76.47% and a specificity of 95.56%. The overall accuracy of the CNN-CAD system was measured at 89.2%.

Notably, the CNN-CAD system exhibited a significantly higher accuracy (by 17.3%) and specificity (by 32.2%) when compared to human endoscopists.

Dehua Tang et al. [34] differentiated between intramucosal and advanced gastric cancer using a deep convolutional neural network (DCNN). The DCNN model exhibited notable discrimination between intramucosal gastric cancer (GC) and advanced GC, as indicated by an area under the curve (AUC) of 0.942. The model achieved a sensitivity of 90.5% and a specificity of 85.3%. When comparing the diagnostic performance of novice and expert endoscopists, it was observed that their accuracy (84.6% vs. 85.5%), sensitivity (85.7% vs. 87.4%), and specificity (83.3% vs. 83.0%) were all similar when assisted by the DCNN model. Furthermore, the mean pairwise kappa (a measure of agreement) among endoscopists significantly increased with the assistance of the DCNN model, improving from 0.430–0.629 to 0.660–0.861. The utilization of the DCNN model was also associated with a reduction in diagnostic duration, from 4.4 s to 3.0 s. Furthermore, the correlation between the perseverance of effort and diagnostic accuracy among endoscopists diminished with the use of the DCNN model, with the correlation coefficient decreasing from 0.470 to 0.076. This suggests that the DCNN model’s assistance helps mitigate the influence of subjective factors, such as perseverance, on diagnostic accuracy.

In a recent study, Gong EJ et al. [18] used 5017 endoscopic images to train two models: one for lesion detection and another for lesion classification in the stomach. In a randomized pilot study, the lesion detection rate for the computer decision support system (CDSS) was 95.6% in internal tests. Although not statistically significant, CDSS-assisted endoscopy showed a higher lesion detection rate (2.0%) compared to conventional screening (1.3%). In a prospective multicenter external test, the CDSS achieved an 81.5% accuracy rate for four-class lesion classification and an 86.4% accuracy rate for invasion depth prediction.

Goto A et al. [14] aimed at differentiating between intramucosal and submucosal early gastric cancers. The performance of this AI classifier was then compared to a majority vote by endoscopists. The results demonstrated that the AI classifier had better internal evaluation scores, with an accuracy of 77%, sensitivity of 76%, specificity of 78%, and an F1 measure of 0.768. Notably, the F1 score is a statistical measure used to evaluate the accuracy of a test. It considers both the precision (the number of correct positive results divided by the number of all positive results) and the recall (the number of correct positive results divided by the number of positive results that should have been identified). The F1 score is the harmonic mean of precision and recall, providing a balance between them. It is especially useful in situations where the class distribution is imbalanced.

In contrast, the endoscopists had lower performance measures, with 72.6% accuracy, 53.6% sensitivity, 91.6% specificity, and an F1 measure of 0.662. Importantly, the diagnostic accuracy improved when the AI and endoscopists collaborated, achieving 78% accuracy, 76% sensitivity, 80% specificity, and an F1 measure of 0.776 on the test images. The combined approach yielded a higher F1 measure compared to using either the AI or endoscopists individually.

### 3.4. Gastric Neoplasm Segmentation

The ENDOANGEL system demonstrated commendable performance with both capsule endoscopy (CE) images and white-light endoscopy (WLE) images, achieving accuracy rates of 85.7% and 88.9%, respectively, when compared to the manual markers labeled by experts. Notably, these results were obtained using an overlap ratio threshold of 0.60 [35]. In the case of endoscopic submucosal dissection (ESD) videos, the resection margins predicted by ENDOANGEL effectively encompassed all areas characterized by high-grade intraepithelial neoplasia and cancers. The minimum distance between the predicted margins and the histological cancer boundary was measured at 3.44 ± 1.45 mm, outperforming the resection margin based on ME-NBI.

Wenju Du et al. [36] validated the use of correlation information among gastroscopic images as a means to enhance the accuracy of EGC segmentation. The authors used the Jaccard similarity index and the Dice similarity coefficient for validation. The Jaccard similarity index is a statistic that measures the similarity and diversity of sample sets. It is defined as the size of the intersection divided by the size of the union of the sample sets. The Jaccard coefficient is widely used in computer science, ecology, genomics, and other sciences where binary or binarized data are used. It is a ratio of Intersection over Union, taking values between 0 and 1, where 0 indicates no similarity and 1 indicates complete similarity.

Similar to the Jaccard index, the Dice similarity coefficient is a measure of the overlap between two samples. For image segmentation, it is calculated as twice the area of overlap between the predicted segmentation and the ground truth divided by the sum of the areas of the predicted segmentation and the ground truth. It is a common metric for evaluating the performance of image segmentation algorithms, particularly in medical imaging.

When applied to an unseen dataset, the EGC segmentation method demonstrated a Jaccard similarity index (JSI) of 84.54%, a threshold Jaccard index (TJI) of 81.73%, a Dice similarity coefficient (DSC) of 91.08%, and a pixel-wise accuracy (PA) of 91.18%. Tingsheng Ling et al. [37] achieved a high accuracy of 83.3% when correctly predicting the differentiation status of EGCs. Notably, in the man–machine contest, the CNN1 model significantly outperformed the five human experts, achieving an accuracy of 86.2% compared to the experts’ accuracy of 69.7%. When delineating EGC margins while utilizing an overlap ratio of 0.80, the model achieved high levels of accuracy for both differentiated EGCs and undifferentiated EGCs, with accuracies of 82.7% and 88.1%, respectively.

For invasive gastric cancer detection and segmentation, Atsushi Teramoto et al. [38] reported 97.0% sensitivity and 99.4% specificity using U-Net. In a case-based evaluation, the approach achieved flawless sensitivity and specificity scores of 100%.

### 3.5. Pre-Malignant Lesion Detection, Classification, and Segmentation

Several studies used AI to detect and classify gastric lesions associated with an increased risk of progression to cancer. A CNN with TResNet [39] was used to classify atrophic gastritis and gastric intestinal metaplasia (GIM). The AUC for atrophic gastritis classification was determined to be 0.98 CI, indicating high discriminatory power. The sensitivity, specificity, and accuracy rates for atrophic gastritis classification were 96.2%, 96.4%, and 96.4%, respectively. Similarly, the AUC for GIM classification was 0.99 (95% CI 0.98–1.00), indicating excellent discriminatory ability. The sensitivity, specificity, and accuracy rates for GIM diagnosis were 97.9%, 97.5%, and 97.6%, respectively. ENDOANGEL-LD [40] achieved 96.9% sensitivity for pre-malignant lesion detection.

Hirai K et al. [41] investigated the yield of CNN in classifying subepithelial lesions in endoscopic ultrasonography (EUS) images. The results showed that the AI system exhibited a commendable 86.1% accuracy when classifying lesions into five distinct categories: Gastrointestinal Stromal Tumors (GIST), leiomyoma, schwannoma, Neuroendocrine Tumors (NET), and ectopic pancreas. Notably, this accuracy rate surpassed that achieved by all endoscopists by a significant margin. Furthermore, the AI system demonstrated a sensitivity of 98.8%, specificity of 67.6%, and an overall accuracy of 89.3% in distinguishing GISTs from non-GISTs, vastly surpassing the sensitivity and accuracy of all endoscopists in the study at the expense of slightly decreased specificity.

Lastly, Passin Pornvoraphat et al. [17] aimed to create a real-time GIM segmentation system using a BiSeNet-based model. This model was able to process images at a rate of 173 frames per second (FPS). The system achieved sensitivity, specificity, positive predictive value (PPV), negative predictive value (NPV), accuracy, and mean intersection over union (IoU) values of 91%, 96%, 91%, 91%, 96%, and 55%, respectively.

### 3.6. Early Gastric Cancer Detection

In recent years, the early detection of GC has gained significant attention [42,43,44,45,46,47,48,49,50,51]. Sakai Y et al. [42] employed transfer learning techniques and achieved a commendable accuracy of 87.6% in detecting EGC. Similarly, Lan Li et al. [43] used Inception-v3 (CNN model), reporting a high sensitivity of 91.2% and a specificity of 90.6%. Tang D et al. [44] evaluated DCNNs and their system demonstrated accuracy rates ranging from 85.1% to 91.2%.

Hu H et al. [12] developed an EGC model based on the VGG-19 architecture. The model was found to be comparable to senior endoscopists in performance but showed significant improvements over junior endoscopists. Wu L et al. [45] conducted a comprehensive study, reporting a sensitivity rate of 87.8% for detecting EGCs, thereby outperforming human endoscopists.

HE X et al. [46] developed ENDOANGEL-ME, which achieved a diagnostic accuracy of 88.4% on internal images and 90.5% on external images. Yao Z et al. [47] introduced a system called EGC-YOLO, achieving accuracy rates of 85.2% and 86.2% in two different test sets. Li J et al. [48] developed another system called ENDOANGEL-LA that significantly outperformed single deep learning models and was on par with expert endoscopists.

Jin J et al. [49] used Mask R-CNN technology and showed high accuracy and sensitivity in both white light images (WLIs) and NBIs. Another noteworthy study [50] utilized the EfficientDet architecture within CNN and achieved a sensitivity of 98.4% and an accuracy of 88.3% in detecting EGC. Finally, Su X et al. [51] assessed three typical Region-based Convolutional Neural Network models, each showing similar accuracy rates, although Cascade RCNN showed a slightly higher specificity of 94.6%.

Figure 2 shows the typical morphological features of malignant (A) and benign (B) ulcers used for AI application.

## 4. Discussion

AI-assisted endoscopy promises to improve the detection rates of pre-malignant and early gastric neoplasias. The addition of computationally efficient DL computer vision models allows endoscopists to identify more potentially treatable gastric lesions during real-time upper endoscopy [15,16,17,18]. AI also shows promise in diagnosing and managing GC, as supported by two recent systematic reviews [8,9]. Our review delves into the use of deep learning (DL) models to detect gastric pre-malignant, early-stage, and neoplastic lesions. This review explores the potential benefits and implications of employing advanced AI techniques in this context.

DL models like Inception, ResNet, and VGGNet demonstrated high accuracy in identifying gastric pathologies [54,55]. However, their real-world performance depends on the quality and diversity of the training data. Training these models on diverse datasets is essential to account for patient demographic factors and subtle differences in disease presentation. AI systems like SSD-GPNet and YOLOv4 are good at detecting gastric pathologies, particularly polyps [24,25,56,57]. However, their real-world effectiveness remains uncertain, as variables like endoscopy lighting conditions, image quality, and inter-operator variability can impact performance.

As a group, the studies reviewed in this analysis have consistently demonstrated promising results in utilizing AI models for GC detection and segmentation. The accuracy achieved by the addition of these AI systems to standard endoscopy has shown significant improvements over un-assisted endoscopy, even outperforming human experts in some cases. The use of CNNs and other DL architectures has proven to be particularly effective in handling the complexity and variability of GC images [19,52,53,58].

In terms of GC detection, the AI models have exhibited high sensitivity and specificity, with values ranging between 80 and 92% in different studies [26,27,28,29,30]. These results suggest that AI has the potential to serve as a reliable tool for the initial screening and detection of GC.

Furthermore, AI-based real-time segmentation systems have shown great promise in segmenting the boundaries of gastric tumors. The segmentation accuracy achieved by these models has consistently surpassed that of traditional endoscopist-based approaches, providing reproducibly and accurately delineated tumor margins [18,36]. This capability is crucial for resection planning, treatment evaluation, and follow-up assessments. The high performance of AI models in differentiating between differentiated and undifferentiated GCs further highlights their potential in guiding treatment decisions and predicting prognoses [49,50,51,52].

Notably, the use of AI-assisted endoscopy significantly improved diagnostic performance and reduced diagnostic variability of endoscopic examinations of the stomach. This was particularly true for less-experienced practitioners. The AI-assisted diagnostic process not only enhanced the sensitivity and specificity of gastric lesion diagnosis, but also reduced the diagnostic duration and enhanced interobserver agreement among endoscopists.

Future ventures for algorithms can be to process and understand a wide range of medical data, from patient records and medical images to genetic data and biomarker profiles. This vast data set can provide AI models a deep understanding of GC patterns. This may allow the models to identify subtle signs and predictive markers that may be missed by human analysis [8,9,10].

AI has shown potential for early GC detection, as shown in multiple studies [42,43,44,45,46,47,48,49,50,51]. Improved endoscopic detection rates of pre-malignant gastric lesions and early GC could make large-scale screening more effective. Using a screening-based approach, earlier diagnosis of premalignant gastric lesions and gastric neoplasias would allow a greater proportion of patients to be treated with endoscopic interventions that have high cure rates and low complication rates [8,51]. Further studies are necessary to assess the feasibility and structure of such screening programs across different populations.

Despite these promising results, there are still challenges that need to be addressed before widespread clinical implementation of AI-assisted endoscopy for pre-malignant and early-stage GC detection and segmentation. One such challenge is the need for larger and more diverse datasets to ensure robust model performance across different populations and imaging conditions. Additionally, there is a requirement for rigorous validation and standardization of AI models to ensure their reliability and generalizability.

Another challenge is the explainability of DL models that plays a pivotal role in clinical acceptance and decision-making. These DL models, trained on vast datasets of endoscopic images, have demonstrated high proficiency in identifying early signs of gastric cancer. However, the challenge lies in deciphering how these models arrive at their conclusions. For gastroenterologists, understanding the rationale behind a model’s prediction is crucial for diagnosis and treatment planning. Explainability in this context involves the model highlighting specific features in endoscopic images, such as subtle changes in tissue texture, color, or vascular patterns, that signify pre-malignant or malignant changes. Techniques like Gradient-weighted Class Activation Mapping (Grad-CAM) [53] can be employed to visualize these decisive image regions, providing clinicians with a visual explanation of the model’s diagnostic process. This level of transparency is essential, as it not only builds trust in the AI system but also aids in the educational aspect, helping clinicians to identify and understand subtle early signs of gastric cancer. Thus, enhancing the explainability of DL models in AI-assisted endoscopy is a crucial step towards integrating these advanced tools into routine clinical practice for more effective and early detection of gastric cancer.

The AI models explored in this review echo the promise of a time when early cancer detection is common, not exceptional. Thus, AI may be a transformative force in the prevention and treatment of early-stage GC in the near future.

In AI-assisted endoscopy for gastric neoplasia, researchers could focus on developing algorithms for enhanced detection and classification of early-stage lesions, offering real-time procedural assistance to endoscopists, and automating pathology correlation to potentially reduce biopsy needs. Further exploration could involve patient-specific risk assessments incorporating personal and genetic data, developing AI-driven post-endoscopic surveillance recommendations, and creating training tools for endoscopists. Additionally, the integration of endoscopic data with other imaging modalities like CT or MRI could provide a more comprehensive diagnostic approach to gastric neoplasia.

We acknowledge the limitations of this review. Most of the included studies were single-center, potentially limiting their scope of patient demographics and disease presentations. Additionally, AI model training and validation often used undisclosed datasets, raising generalizability concerns. The retrospective training and validation of AI algorithms may not fully reflect prospective clinical performance. Furthermore, the lack of disclosure of models’ technical details can make study reproduction difficult [53,59,60,61]. Lastly, we selected PubMed/MEDLINE for its relevance in biomedical research, recognizing that this choice narrows our review’s scope. This might exclude studies from other databases, possibly limiting diverse insights. This limitation was considered to balance focus and comprehensiveness in our review.

In conclusion, AI’s potential in diagnosing and managing pre-malignant early-stage gastric neoplasia is significant due to several reasons. Firstly, AI algorithms, powered by machine learning, can analyze complex medical data at unprecedented speeds and with high accuracy. This ability enables the early detection of gastric neoplasia, which is crucial for successful treatment outcomes.

Secondly, AI can assist in differentiating between benign and malignant gastric lesions, which is often a challenging task in medical practice. By doing so, AI can help reduce unnecessary biopsies and surgeries, leading to better patient outcomes and reduced healthcare costs.

Thirdly, AI can continuously learn from new data, enhancing its diagnostic capabilities over time. This aspect is particularly important in the field of oncology, where early and accurate diagnosis can significantly impact patient survival rates.

However, further research, larger-scale studies, and prospective clinical trials are necessary to confirm these findings. Future research should also focus on transparency, providing detailed reports on models’ technical details and the datasets used for training and validation.

## Figures and Tables

**Figure 1 diagnostics-13-03613-f001:**
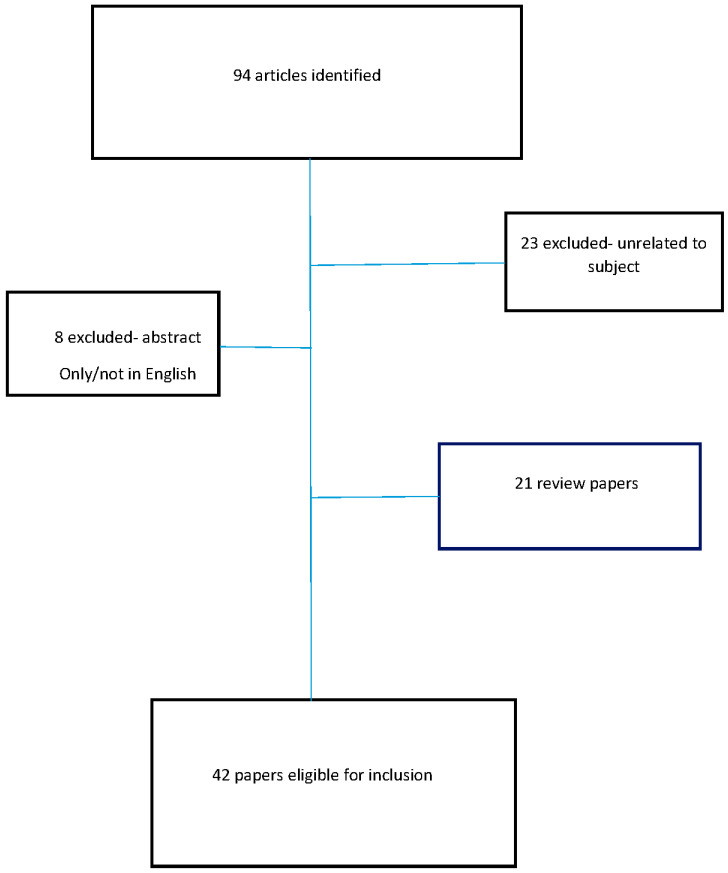
Flowchart delineating the selection procedure of the studies included in this review.

**Figure 2 diagnostics-13-03613-f002:**
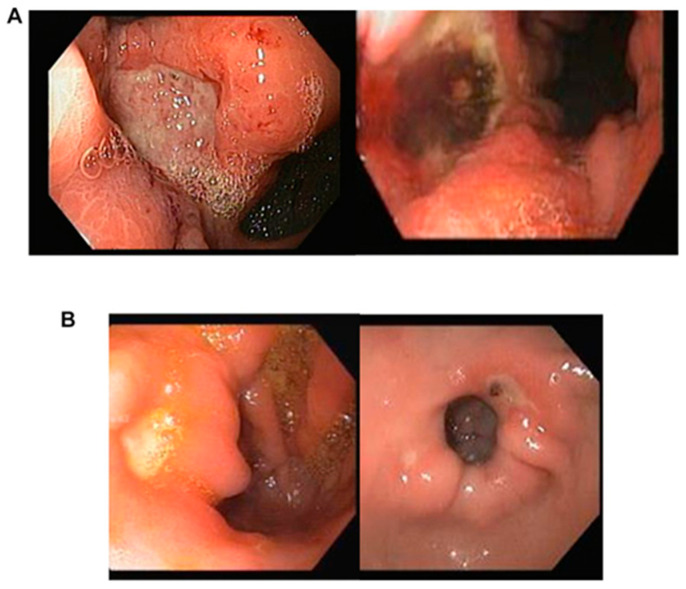
Morphological attributes of malignant (**A**) and benign (**B**) ulcers are as follows: (**A**) Malignant gastric ulcers exhibit typically larger dimensions, measuring over 1 cm, and possess irregular borders. These borders display elevation in comparison to the ulcer base, and discoloration of the base is frequently observed. (**B**) Conversely, benign gastric ulcers tend to be smaller in size, measuring less than 1 cm, and feature a clean unblemished base with consistently flat and regular borders.

**Table 1 diagnostics-13-03613-t001:** Research characteristics of all studies included in this review.

REF	Subject	Clinical Task	Study Design	Sample Size	Detection/Classification Objective	AI Model/Algorithm	AI Model Performance	Clinical Implications
Liu L et al. [11]	Pre-malignant lesions	Diagnosis and segmentation of GNLs	Retrospective, two centers	3757 images from 392 patients with GNLs and 2420 images from 568 patients with non-GNLs	Diagnosis and segmentation of GNLs under magnifying endoscopy with narrow-band imaging (ME-NBI)	Two convolutional neural network (CNN) modules	Accuracy: 90.8%, Sensitivity: 92.5%, Specificity: 89.0%	CAD system can assist endoscopists in more accurately diagnosing GNLs and delineating their extent
Hu H et al. [12]	Early gastric cancer	Diagnosis of EGC under ME-NBI	Multicenter, randomized	295 cases	Diagnosis of EGC	VGG-19	AUC: 0.808 (ITC), 0.813 (ETC), Accuracy: 0.770, Sensitivity: 0.792, Specificity: 0.745	The model exhibited comparable performance with senior endoscopists in the diagnosis of EGC and showed potential value in aiding and improving the diagnosis of EGC by endoscopists.
Zhu Y et al. [13]	Gastric cancer	Prediction of invasion depth	Retrospective and prospective	790 images in the development dataset, 203 images in the test dataset	Determination of invasion depth of gastric cancer	ResNet50	Area under the ROC curve: 0.94	Assists in screening patients for endoscopic resection by accurately predicting invasion depth.
Goto A et al. [14]	Early gastric cancer	Invasion depth determination	Retrospective single	250 intramucosal cancers and 250 submucosal cancers	Differentiating intramucosal and submucosal gastric cancers	Not specified	Accuracy: 77%, Sensitivity: 76%, Specificity: 78%, F1 measure: 0.768	Improvement in diagnostic ability to determine invasion depth of early gastric cancer
DU H et al. [16]	Detection of gastric pathologies	Real-time diagnosis	Retrospective and prospective	4201 images, 7436 image-pairs, 162 videos	Real-time diagnosing of gastric neoplasms	ENDOANGEL-MM	Accuracy: 86.54% for images, 90.00% for videos, 93.55% for prospective patients	ENDOANGEL-MM identifies gastric neoplasms with good accuracy and has potential role in real-clinic
Pornvoraphat P et al. [17]	Pre-malignant lesions	Real-time segmentation of GIM	Retrospective, single-center, case–control	940 GIM images and 1239 non-GIM images	Segmentation of GIM from a healthy stomach	BiSeNet-based model	Sensitivity: 91%, Specificity: 96%, Accuracy: 96%, Mean IoU: 55%	Real-time detection of GIM for improved diagnostic precision
Gong EG et al. [18]	Detection of gastric pathologies	Automated detection and classification of gastric neoplasms	Prospective, multicenter	5017 images for training, 2524 procedures for internal testing, 3976 images from five institutions for external testing	Automated detection and classification of gastric neoplasms in real-time endoscopy	Clinical decision support system (CDSS) based on deep learning	Detection rate: 95.6% (internal test), Accuracy: 81.5% in four-class classification and 86.4% in binary classification (external test)	CDSS has potential for real-life clinical application and high performance in terms of lesion detection and classification
Lee JH et al. [19]	Classification of gastric pathologies	Classification of benign ulcer and cancer	Retrospective, single center, case control	200 normal, 367 cancer, and 220 ulcer cases	Classification of normal, benign ulcer, and cancer images	Inception, ResNet, and VGGNet	AUC for the three classifiers: 0.95, 0.97, and 0.85, respectively	Automatic classification can complement manual inspection efforts to minimize risks of missing positives due to repetitive sequence of endoscopic frames.
Cho BJ et al. [20]	Classification of gastric pathologies	Classification of gastric neoplasms	Retrospective and prospective case and control, two centers	5017 images from 1269 patients for training, 812 images from 212 patients for testing, and 200 images from 200 patients for validation	Automatic classification of gastric neoplasms	Inception-Resnet-v2	Weighted average accuracy: 84.6%; AUC for differentiating gastric cancer and neoplasm: 0.877 and 0.927, respectively	Potentially useful for clinical application in classifying gastric cancer or neoplasm using endoscopic white-light images.
Ueyama H et al. [21]	Gastric cancer	Diagnosis	Retrospective and prospective	5574 ME-NBI images (3797 EGCs, 1777 non-cancerous)	Diagnosis of early gastric cancer (EGC)	ResNet50	Accuracy: 98.7%, Sensitivity: 98%, Specificity: 100%	The AI-assisted CNN-CAD system for ME-NBI diagnosis of EGC could process many stored ME-NBI images in a short period of time and had a high diagnostic ability. This system may have great potential for future application to real clinical settings.
Horiuchi Y et al. [22]	Early gastric cancer	Differentiating EGC from gastritis	Retrospective and prospective single	pre-trained using 1492 EGC and 1078 gastritis images from ME-NBI. A separate test dataset (151 EGC and 107 gastritis images based on ME-NBI) was used to evaluate the diagnostic ability	Differentiating EGC from gastritis	22-layer CNN	Accuracy: 85.3%, Sensitivity: 95.4%, Specificity: 71.0%	May provide rapid and sensitive differentiation between EGC and gastritis
Klang E et al. [23]	Gastric ulcers	Malignancy detection	Retrospective	1978 GU images	Discrimination between benign and malignant gastric ulcers	CNN	AUC: 0.91, Sensitivity: 92%, Specificity: 75%	The algorithm may improve the accuracy of differentiating benign from malignant ulcers during endoscopies and assist in patients’ stratification, allowing accelerated patient management and an individualized approach towards surveillance endoscopy.
Liu Y et al. [24]	Gastric ulcers	Classification of benign and malignant gastric ulcer lesions	Retrospective, single-center, case–control	109 cases in the benign group, 69 in malignant	Automatic classification and diagnosis of benign and malignant gastric ulcer lesions	Convolutional neural network, Xception model with residual attention module	Accuracy: 81.411%, F1 score: 81.815%, Sensitivity: 83.751%, Specificity: 76.827%, Precision: 80.111%	Residual attention mechanism can improve the classification effect of Xception CNN on benign and malignant lesions of gastric ulcers
Zhang X et al. [25]	Detection of gastric pathologies	Polyp detection	Single center, retrospective, cases	404 cases	Automatic detection of gastric polyps	Single Shot MultiBox Detector (SSD-GPNet)	mAP: 90.4%; Improved polyp detection recall by over 10%	Assists in reducing gastric polyp miss rate and potentially decreases the burden on physicians.
Durak S et al. [26]	Detection of gastric pathologies	Detection of gastric polyps	Retrospective study	2195 endoscopic images and 3031 polyp labels	Automatic gastric polyp detection	YOLOv4, CenterNet, EfficientNet, Cross Stage ResNext50-SPP, YOLOv3, YOLOv3-SPP, Single Shot Detection, Faster Regional CNN	YOLOv4 had 87.95% mean average precision	YOLOv4 can be used effectively in gastrointestinal CAD systems for polyp detection
Wu L et al. [27]	Detection of gastric pathologies	Detection of gastric neoplasms	Single-center, randomized controlled, tandem trial	1812 patients	AI-assisted detection of gastric neoplasms	Not specified	Lower gastric neoplasm miss rate in the AI-first group	AI can reduce the miss rate of gastric neoplasms in clinical practice
Xu M et al. [28]	Detection of gastric pathologies	Detecting gastric precancerous conditions	Retrospective, multicenter	760 patients	Detecting gastric atrophy (GA) and intestinal metaplasia (IM)	ENDOANGEL (a deep CNN)	Diagnostic accuracy of GA was 0.901	The model shows potential for real-time detection of gastric precancerous conditions in clinical practice.
Hirasawa T et al. [29]	Gastric cancer	Gastric cancer detection	Retrospective and prospective	69 patients with 77 gastric cancer lesions. Trained using 13,584 endoscopic images of gastric cancer. To evaluate the diagnostic accuracy, an independent test set of 2296 stomach images collected from 69 consecutive patients with 77 gastric cancer lesions was applied to the constructed CNN.	Automatic detection of gastric cancer	Single Shot MultiBox Detector (SSD)	Sensitivity: 92.2%; Positive Predictive Value: 30.6%	Reduces the burden on endoscopists by processing numerous stored endoscopic images in a very short time.
Ikenoyama Y et al. [30]	Gastric cancer	Early detection	Retrospective and prospective single	13,584 images from 2639 lesions. The CNN was constructed using 13,584 endoscopic images from 2639 lesions of gastric cancer. Subsequently, its diagnostic ability was compared to that of 67 endoscopists using an independent test dataset (2940 images from 140 cases).	Detecting early gastric cancer	CNN	Sensitivity: 58.4%, Specificity: 87.3%	The CNN detected more early gastric cancer cases in a shorter time than the endoscopists. A diagnostic support tool for gastric cancer using a CNN will be realized in the near future.
Bang CS et al. [31]	Classification of gastric pathologies	Classify the invasion depth of gastric neoplasms	Prospective, multicenter	a total of 5017 images from 1269 individuals; of these, 812 images from 212 subjects were used	Classifying the invasion depth of gastric neoplasms	AutoDL models (Neuro-T, Create ML Image Classifier, AutoML Vision)	Accuracy: 89.3%	AutoDL models showed high accuracy in classifying invasion depth of gastric neoplasms, suggesting their potential for improving diagnostic accuracy and efficiency in clinical practice.
Yoon HJ et al. [32]	Early gastric cancer	Tumor invasion depth prediction	Retrospective single	11,539 images	Classification of endoscopic images as EGC or non-EGC	VGG-16	AUC for EGC detection: 0.981, AUC for depth prediction: 0.851	May improve depth prediction in EGC, particularly in undifferentiated-type histology, requiring further validation
Hamada K et al. [33]	Early gastric cancer	Evaluating the depth of invasion of early gastric cancer	Retrospective	200 cases	Evaluating the depth of invasion of early gastric cancer	ResNet152	Sensitivity, specificity, and accuracy for diagnosing M cancer were 84.9%	Assist in endoscopic diagnosis of early gastric cancer
Tang D et al. [34]	Gastric cancer	Diagnosis of intramucosal GC	Retrospective single-center	666 gastric cancer patients 3407 endoscopic images	Discrimination of intramucosal GC from advanced GC	DCNN	AUC: 0.942	The model achieved high accuracy in discriminating intramucosal GC from advanced GC, indicating its potential to assist endoscopists in diagnosing intramucosal GC.
An P et al. [35]	Early gastric cancer	Delineation of cancer margins	Retrospective and prospective	A total of 546 CE images from 67 patients were included, and 34 CE images from 14 patients were included in the test dataset. In the WLE dataset, the training dataset consisted of 343 images from 260 patients, and the test dataset consisted of 321 images from 218 patients	Delineating the resection margin of early gastric cancer	Fully convolutional networks (ENDOANGEL)	Accuracy: 85.7% in CE images, 88.9% in WLE images	Assisting endoscopists in delineating the resection extent of EGC during ESD
Du W et al. [36]	Early gastric cancer	Automatic segmentation of EGC lesions	Retrospective and prospective	7169 images from 2480 patients	Segmentation of EGC lesions in gastroscopic images	Co-spatial attention and channel attention-based triple-branch ResUnet (CSA-CA-TB-ResUnet)	Jaccard similarity index (JSI) of 84.54%	Accurate segmentation of EGC lesions for aiding clinical diagnosis and treatment
Ling T et al. [37]	Gastric cancer	Margin delineation	Retrospective	2217 images from 145 EGC patients, 1870 images from 139 EGC patients	Identification of differentiation status and delineation of margins of EGC	CNN	Accuracy: 83.3%	The AI system accurately identifies the differentiation status of EGCs and may assist in determining the surgical strategy and achieving curative resection in EGC patients.
Teramoto A et al. [38]	Gastric cancer	Detection and segmentation of invasive gastric cancer	Retrospective single-center	2378 images from different patients	Classification of endoscopic images and identification of the extent of cancer invasion	Cascaded deep learning model, U-Net	Sensitivity: 97.0%, Specificity: 99.4%, Case-based evaluation: 100%	The method could be useful for the classification of endoscopic images and identification of the extent of cancer invasion
Lin N et al. [39]	Pre-malignant lesions	Recognition of AG and GIM	Retrospective multicenter	A total of 7037 endoscopic images from 2741 participants were used to develop the CNN	Simultaneous recognition of AG and GIM	Deep Convolutional Neural Network (CNN) using TResNet	Not specified	CNN model could be used for diagnosing AG and GIM
Wu L et al. [40]	Early gastric cancer	Detecting gastric neoplasm, identifying EGC, and predicting EGC invasion depth and differentiation status	Multicenter, prospective, real-time, competitive comparative, diagnostic study	100 videos	Detecting neoplasms and diagnosing EGCs	Not specified	Sensitivity rates of the system for detecting neoplasms and diagnosing EGCs were 87.81% and 100%, respectively	AI system can enhance the performance of endoscopists in diagnosing EGC
Hirai K et al. [41]	Pre-malignant lesions	Differentiating gastrointestinal stromal tumors from benign subepithelial lesions	Retrospective	631 cases	Classifying SELs on EUS images	CNN	Accuracy of 86.1% for five-category classification, sensitivity and accuracy of 98.8% and 89.3%, respectively, for differentiating GISTs from non-GISTs	Assist in improving the diagnosis of SELs in clinical practice
Sakai Y et al. [42]	Early gastric cancer	Early gastric cancer detection	Retrospective single center	1000 images	Automatic detection of early gastric cancer	Not provided	Accuracy: 87.6%; Balanced sensitivity and specificity	Assists endoscopists in decision-making by providing a heat map of candidate regions of early gastric cancer.
Li L et al. [43]	Early gastric cancer	Diagnosis of early gastric cancer	Retrospective and prospective	A total of 386 images of non-cancerous lesions and 1702 images of early gastric cancer were collected to train and establish a CNN model (Inception-v3). Then, a total of 341 endoscopic images (171 non-cancerous lesions and 170 early gastric cancer) were selected to evaluate the diagnostic capabilities of CNN and endoscopists.	Diagnosis of early gastric cancer	Inception-v3	Sensitivity: 91.18%, Specificity: 90.64%, Accuracy: 90.91%	May enhance the diagnostic efficacy of non-experts in differentiating early gastric cancer from non-cancerous lesions
Tang D et al. [44]	Early gastric cancer	Detection of early gastric cancer	Retrospective and prospective, multicenter	All 45,240 endoscopic images from 1364 patients were divided into a training dataset (35,823 images from 1085 patients) and a validation dataset (9417 images from 279 patients). Another 1514 images from three other hospitals were used as external validation.	Detection of early gastric cancer	Deep Convolutional Neural Networks (DCNNs)	Accuracy: 85.1%–91.2%, Sensitivity: 85.9%–95.5%, Specificity: 81.7%–90.3%	Multicenter prospective validation needed for clinical application
Wu L et al. [45]	Pre-malignant lesions	Detecting gastric lesions and predicting neoplasms	Retrospective and prospective	Over 10,000	Assisting in detecting gastric lesions and predicting neoplasms by WLE	ENDOANGEL-LD	Sensitivity of 96.9% for detecting gastric lesions and 92.9% for diagnosing neoplasms in internal patients	Assisting endoscopists in screening gastric lesions and suspicious neoplasms in clinical work
He X et al. [46]	Early gastric cancer	Diagnosing EGC in magnifying image-enhanced endoscopy	Retrospective and prospective	3099 cases	Diagnosing EGC in M-IEE	ENDOANGEL-ME	Diagnostic accuracy of 88.44% and 90.49% in internal and external images, respectively	Assist in diagnosing early gastric cancer
Yao Z et al. [47]	Early gastric cancer	Diagnosing early gastric cancer	Prospective	1653 cases	Rapid and accurate diagnosis of endoscopic images from early gastric cancer	YOLO	Accuracy, sensitivity, specificity, and positive predictive value of 85.15%, 85.36%, 84.41%, and 95.22%, respectively, for Test Set 1	Assist in the efficient, accurate, and rapid detection of early gastric cancer lesions
Li J et al. [48]	Early gastric cancer	Diagnosis of EGC under M-IEE	Retrospective	692 patients	Develop a logical anthropomorphic AI diagnostic system for EGCs under M-IEE	ENDOANGEL-LA, based on feature extraction, deep learning (DL), and machine learning (ML)	Accuracy of ENDOANGEL-LA in images was 88.76% and in videos was 87.00%	ENDOANGEL-LA has the potential to increase interactivity between endoscopists and CADs, and improve trust and acceptability of CADs for endoscopists
Jin Z et al. [49]	Early gastric cancer	Automatic detection of early gastric cancer	Controlled trials	7133 images from different patients	Automatic detection of early gastric cancer	Mask R-CNN	WLI test—Accuracy: 90.25%, Sensitivity: 91.06%, Specificity: 89.01%, NBI test—Accuracy: 95.12%, Sensitivity: 97.59%	Can be effectively applied to clinical settings for the detection of EGC, especially for the real-time analysis of WLIs
Zhou B et al. [50]	Early gastric cancer	Detection of early gastric cancer	Single-center retrospective study	5770 images from 194 patients	Automatic detection of early gastric cancer	EfficientDet	Case-based Sensitivity: 98.4%, Image-based Accuracy: 88.3%, Sensitivity: 84.5%, Specificity: 90.5%	Shows great potential in assisting endoscopists with the detection of EGC
Su X et al. [51]	Early gastric cancer	EGC detection	Retrospective single-center	3659 cases	Detection of early gastric cancer	Faster RCNN, Cascade RCNN, Mask RCNN	Accuracy: 0.935 (Faster RCNN), 0.938 (Cascade RCNN), 0.935 (Mask RCNN), Specificity: 0.908 (Faster RCNN, Mask RCNN), 0.946 (Cascade RCNN)	These deep learning methods can assist in early gastric cancer diagnosis using endoscopic images.
Nakahira H et al. [52]	Gastric cancer	Risk stratification of GC	Retrospective single-center	107,284 images	Stratification of GC risk	Not explicitly stated	Not explicitly stated	Could provide effective surveillance for GC, stratifying GC risk based on endoscopic examinations
Igarashi S et al. [53]	Gastric cancer	Automated localization of digestive lesions and prediction of cancer invasion depth	Retrospective	441 patients	Classification of upper GI organ images into anatomical categories	AlexNet	Accuracy: Training dataset 0.993, Validation dataset 0.965	Facilitating data collection and assessment of EGD images, potentially useful for both expert and non-expert endoscopists

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
