# Peer review of "Deep Learning and Gastric Cancer: Systematic Review of AI-Assisted Endoscopy"

_diagnostics, 2023, doi:10.3390/diagnostics13243613_

Round 1

Reviewer 1 Report

Comments and Suggestions for Authors

Dear Authors,

I read with great interest your manuscript about deep learning and the innovative trends in use of  artificial intelligence in medicine, especially in endoscopy. I appreciate the work and  the effort to summarize the existent literature on this hot subject. 

I find it will be useful to consider some  explanations to be added:

-in the introduction the principle of deep learning and the fact that there are so many AI tools.

-also, I find it useful to explain the segmentation process ( as mapping ).

-the F1 score , jaccard similarity index , Dice similarity coefficient to add a short explanation

-to mention in the first  paragraph of Gastric neoplasm detection to mention that SSD-GPNet was shown to significantly increase gastric polyp recalls ( it was mentioned just polyp) . (There are also CAD for colonic polyps, and for sure it is deductible that the subject is the stomach in your paper . Still , I consider it should be mentioned). Also, avoid to be redundant with the text and table information. 

I find the paper interesting and congratulation for your work.

Comments on the Quality of English Language

There are some misspelling  errors that should be corrected.

Author Response

Reviewer 1

 read with great interest your manuscript about deep learning and the innovative trends in use of  artificial intelligence in medicine, especially in endoscopy. I appreciate the work and  the effort to summarize the existent literature on this hot subject. 

I find it will be useful to consider some explanations to be added:

-in the introduction the principle of deep learning and the fact that there are so many AI tools.

Dear reviewer, thank you for your suggestion. Accordingly, the following text was added to the introduction:

Deep learning is a sophisticated branch of artificial intelligence that uses algorithms inspired by the human brain, known as artificial neural networks. These networks consist of layers of nodes that transform input data, enabling the machine to learn from large datasets. The learning process involves adjusting the network's internal parameters to reduce prediction errors, utilizing methods like backpropagation and gradient descent. Deep learning is distinguished by its use of numerous layers that enable the recognition of complex patterns, making it ideal for applications in image and speech recognition, among others.The proliferation of AI tools can be largely attributed to advances in computational power, algorithmic improvements, and the abundance of data available in the digital age. These tools have found applications across various sectors, including healthcare, automotive, finance, and customer service. The development of accessible frameworks like TensorFlow and PyTorch has democratized AI, allowing a broader range of users to develop AI models. Additionally, the integration of AI into products and services by many companies has fueled further demand for these tools. ( page  5-6, lines 89-101 , text highlighted).

-also, I find it useful to explain the segmentation process ( as mapping ).

Dear reviewer, thank you for your suggestion. Accordingly, the following text was added to the introduction in order to better explain segmentation process:

Through segmentation process, which begins with collecting and preprocessing gastric images, followed by expert annotation to mark cancerous tissues, the AI model is trained on these annotated images to learn distinguishing features of neoplastic tissue. Once trained, the model can segment neoplastic areas in new images, aiding in diagnosis and treatment planning. ( page 6, lines 104-108 , text highlighted).

-the F1 score , jaccard similarity index , Dice similarity coefficient to add a short explanation

Dear reviewer, thank you for your comment. The following explanations were added to the text:

F1 Score: The F1 score is a statistical measure used to evaluate the accuracy of a test. It considers both the precision (the number of correct positive results divided by the number of all positive results) and the recall (the number of correct positive results divided by the number of positive results that should have been identified). The F1 score is the harmonic mean of precision and recall, providing a balance between them. It's especially useful in situations where the class distribution is imbalanced. ( page 15, lines  286-291, text highlighted).

Jaccard Similarity Index: Also known as the Jaccard index or Intersection over Union, this metric measures the similarity and diversity of sample sets. In the context of image segmentation, it's calculated as the size of the intersection divided by the size of the union of the sample sets. This index is used to evaluate the accuracy of the segmentation by comparing the predicted segmentation to the ground truth.

Dice Similarity Coefficient: Similar to the Jaccard index, the Dice similarity coefficient is a measure of the overlap between two samples. For image segmentation, it's calculated as twice the area of overlap between the predicted segmentation and the ground truth, divided by the sum of the areas of the predicted segmentation and the ground truth. It's a common metric for evaluating the performance of image segmentation algorithms, particularly in medical imaging. ( page 16 lines 308-318, text highlighted).

-to mention in the first  paragraph of Gastric neoplasm detection to mention that SSD-GPNet was shown to significantly increase gastric polyp recalls ( it was mentioned just polyp) . (There are also CAD for colonic polyps, and for sure it is deductible that the subject is the stomach in your paper . Still , I consider it should be mentioned). Also, avoid to be redundant with the text and table information. 

The sentence was in the text:

The SSD-GPNet was shown to significantly increase polyp detection recalls by more than 10% (p < 0.001), particularly in the detection of small polyps ( page 12 lines 217-218, text highlighted)

I find the paper interesting and congratulation for your work.

Thank you :)

Reviewer 2 Report

Comments and Suggestions for Authors

Comments

Abstract

1.       The term "nepolasia" in "gastric nepolasia detection" appears to be a typographical error. It should be corrected to "neoplasia."

2.       The abstract could benefit from a brief mention of the limitations of the review, particularly regarding the heterogeneity in DL algorithms and study designs. This would provide a more comprehensive overview.

Introduction

3.       The introduction could benefit from a brief statement about the increasing prevalence of gastric cancer globally. This would further emphasize the importance of improving detection methods.

Methods

4.       It would be beneficial to briefly mention the inclusion of a specific timeframe for the literature search. This would provide additional context for the selection of studies.

5.       While the data extraction process is well-described, it may be helpful to briefly mention the software or tools used for data extraction to ensure transparency.

Result

6.       Ensure consistent formatting of citations (e.g., Lee JH et al. (15), Cho BJ et al (16)) throughout the section.

7.       For studies involving specific models (e.g., ResNet50, VGG-16), consider briefly explaining the architecture of the model for readers who may not be familiar with them.

8.       Clarify whether the sensitivity, specificity, and accuracy values reported are for the AI models, human endoscopists, or both.

9.       It may be helpful to briefly explain terms such as AUC, mAP, sensitivity, specificity, etc., for readers who may not be familiar with these metrics.

Discussion

10.   When discussing future research directions, consider providing specific examples or areas of focus that researchers could explore in the context of AI-assisted endoscopy for gastric neoplasia.

11.   In the final paragraph, consider rephrasing "AI's potential in diagnosing and managing pre-malignant, early stage and gastric neoplasia is significant" for clarity. Perhaps specify "AI's potential in diagnosing and managing pre-malignant, early-stage gastric neoplasia is significant."

Comments on the Quality of English Language

Moderate editing of English language required

Author Response

Reviewer 2

Abstract

  1. The term "nepolasia" in "gastric nepolasia detection" appears to be a typographical error. It should be corrected to "neoplasia."

Typo was corrected, thank you for your comment .( page 3 line 45, text highlighted)

  1. The abstract could benefit from a brief mention of the limitations of the review, particularly regarding the heterogeneity in DL algorithms and study designs. This would provide a more comprehensive overview.

Dear reviewer, thank you for this comment. The following text was added to the abstract:

This review is limited by predominantly single-center studies and undisclosed datasets in AI training, impacting generalizability and demographic representation. Further, retrospective algorithm training may not reflect actual clinical performance, and a lack of model details hinders replication efforts. ( page 4 lines 58-61, text highlighted )

Introduction

  1. The introduction could benefit from a brief statement about the increasing prevalence of gastric cancer globally. This would further emphasize the importance of improving detection methods.

Dear reviewer, according to your comment the following text was added to the introduction:

The global prevalence of gastric cancer is increasing significantly, emphasizing the need for improved detection methods. Currently, gastric cancer ranks as the fifth most common malignant cancer and the fourth leading cause of cancer-related mortality worldwide. Despite a decline in incidence rates, the global burden of this malignancy is projected to increase by 62% by 2040 (2) . This escalation is further highlighted by predictions from the International Agency for Research on Cancer (IARC), which forecast an increase to about 1.8 million new cases and approximately 1.3 million deaths by 2040, representing increases of about 63% and 66%, respectively, compared with 2020.(3) ( page 5 lines 70-77)

Methods

  1. It would be beneficial to briefly mention the inclusion of a specific timeframe for the literature search. This would provide additional context for the selection of studies.

Time frame was included in the methods text:

from their inception until May 2023 ( page 8 line 129, text highlighted)

  1. While the data extraction process is well-described, it may be helpful to briefly mention the software or tools used for data extraction to ensure transparency.

Dear reviewer, thank you for your constructive suggestion to enhance transparency in our data extraction process. We agree that mentioning the tools used adds value to our methodology. In response to your comment, we have updated the methods section as follows:

Two reviewers independently extracted data using a pre-defined extraction form in Microsoft Excel. The data included variables like the first author's name, publication year, and study design, among others. Disagreements were resolved through discussion or consultation with a third reviewer. (page 8 , lines 144-146, text highlighted)

Result

  1. Ensure consistent formatting of citations (e.g., Lee JH et al. (15), Cho BJ et al (16)) throughout the section.

Dear reviewer, thank you for your comment. Text was revised and changed accordingly.

  1. For studies involving specific models (e.g., ResNet50, VGG-16), consider briefly explaining the architecture of the model for readers who may not be familiar with them.

Thank you for your comments. The following text was added to the results section:

The ResNet50 model is a variant of the ResNet (Residual Network) architecture, which is widely used in deep learning for image recognition and processing tasks. It has 50 layers deep, hence the '50' in its name. One of the key features of ResNet is the use of "residual blocks" that help in combating the vanishing gradient problem in deep neural networks, enabling the training of much deeper networks.In a ResNet50 model, these residual blocks consist of layers with skip connections that allow the activation from one layer to be fast-forwarded to a later layer, bypassing one or more layers in between. This design helps in preserving the learning and signal from the initial layers to the deeper layers in the network. ResNet50, specifically, is known for its balance between depth and complexity, making it a popular choice for many image recognition tasks.( page 10 lines 180-188  , text highlighted)

VGG-16 is a convolutional neural network model proposed by the Visual Graphics Group (VGG) from the University of Oxford. It is composed of 16 layers (hence the name VGG-16), including 13 convolutional layers and 3 fully connected layers.One of the key characteristics of VGG-16 is its use of a large number of convolutional layers with small-sized filters (3x3), which allows it to capture complex patterns in the data while keeping the computational complexity manageable. ( page 13 lines 246-250)

Notably, Inception, ResNet, and VGGNet are influential architectures in the field of deep learning, particularly for image processing tasks:

Inception: Known for its efficiency and depth, the Inception architecture, especially its popular version Inception v3, uses a combination of different-sized convolutional filters within the same layer (called a "network in network" approach). This allows it to capture spatial hierarchies at different scales and reduces computational cost.

ResNet (Residual Network): This architecture introduced residual learning to alleviate the vanishing gradient problem in very deep networks. By using skip connections that bypass one or more layers, ResNet can effectively train networks with many layers, including popular variants like ResNet50 and esNet101.

VGGNet (Visual Geometry Group Network): VGGNet, particularly VGG16 and VGG19, is known for its deep but simple architecture, using multiple convolutional layers with small-sized filters (3x3) before pooling layers. Its uniform architecture makes it easy to understand and implement, and it has been highly influential in demonstrating the effectiveness of depth in convolutional neural networks.(15) ( page  9-10,  lines 163-176 , text highlighted)

The Xception model is an advanced convolutional neural network (CNN) architecture that builds upon the principles of Inception networks. The key innovation in Xception is the concept of depthwise separable convolutions. This approach separates the convolution process into two parts: a depthwise convolution that applies a single filter per input channel, followed by a pointwise convolution that applies a 1x1 convolution to combine the outputs of the depthwise convolution. (page  11 , lines 195-199, text highlighted)

  1. Clarify whether the sensitivity, specificity, and accuracy values reported are for the AI models, human endoscopists, or both.

Thank you for your comment. The following sentence was added to the results:

All sensitivity, specificity, and accuracy values represent AI models’ performance. ( page 9 line 158, text highlighted )

  1. It may be helpful to briefly explain terms such as AUC, mAP, sensitivity, specificity, etc., for readers who may not be familiar with these metrics.

Discussion

  1. When discussing future research directions, consider providing specific examples or areas of focus that researchers could explore in the context of AI-assisted endoscopy for gastric neoplasia.

Thank you for your comment. The following text was added to the discussion:

In AI-assisted endoscopy for gastric neoplasia, researchers could focus on developing algorithms for enhanced detection and classification of early-stage lesions, offering real-time procedural assistance to endoscopists, and automating pathology correlation to potentially reduce biopsy needs. Further exploration could involve patient-specific risk assessments incorporating personal and genetic data, developing AI-driven post-endoscopic surveillance recommendations, and creating training tools for endoscopists. Additionally, integrating endoscopic data with other imaging modalities like CT or MRI could provide a more comprehensive diagnostic approach to gastric neoplasia.( page 22  lines 446-452, text highlighted)

  1. In the final paragraph, consider rephrasing "AI's potential in diagnosing and managing pre-malignant, early stage and gastric neoplasia is significant" for clarity. Perhaps specify "AI's potential in diagnosing and managing pre-malignant, early-stage gastric neoplasia is significant."

Dear reviewer, thank you for your comment. The sentence was changed according to your suggestion:

AI's potential in diagnosing and managing pre-malignant, early stage gastric neoplasia is significant due to several reasons. Firstly, AI algorithms, powered by machine learning, can analyze complex medical data at unprecedented speeds and with high accuracy. This ability enables the early detection of gastric neoplasia, which is crucial for successful treatment outcomes.Secondly, AI can assist in differentiating between benign and malignant gastric lesions, which is often a challenging task in medical practice. By doing so, AI can help reduce unnecessary biopsies and surgeries, leading to better patient outcomes and reduced healthcare costs.Thirdly, AI can continuously learn from new data, enhancing its diagnostic capabilities over time. This aspect is particularly important in the field of oncology, where early and accurate diagnosis can significantly impact patient survival rates. (page 23 lines 462-471, text highlighted)

Reviewer 3 Report

Comments and Suggestions for Authors

This is a very good review of DL for gastric cancer. However, the authors should consider the following points to improve the paper.

1. The methodology used should be improved. The authors have stated they only considered publications from Pubmed/MEDLINE. The reason for this should be justified. Otherwise, authors should consider other outlets.

2. The problem statement describing GC as an issue should be more elaborate in the introduction section, discussing the demography of the sufferers of the disease and so on. 

3. The review has concentrated on the DL techniques used in the classification, detection, and segmentation of GC. However, the authors could have included the optimal process or general framework used for the individual tasks and do a comparison of the processes. 

4. Comparison of the data types should be done as well including the advantages and disadvantages and this can be linked to the algorithm used for each of the data types. for example, how do DL techniques used for CT scans differ from mammograms? How do they compare in terms of the statistical metrics used for evaluation?

5. Authors should also discuss the issue of explainability of the DL model as there are lots of comparisons to the traditional endoscopist in the review

6. Authors have suggested future work to consider approaches that use multi-modal data for GC analysis. There is already lots of work in this area and authors may want to include this in their review as well.

Author Response

Reviewer 3

This is a very good review of DL for gastric cancer. However, the authors should consider the following points to improve the paper.

  1. The methodology used should be improved. The authors have stated they only considered publications from Pubmed/MEDLINE. The reason for this should be justified. Otherwise, authors should consider other outlets.

Dear reviewer, we greatly appreciate your insightful feedback on our methodology. In our review, we selected PubMed/MEDLINE for its specificity and peer-reviewed content in biomedical literature, aligning with our study's focus. We understand and value your point about the scope of sources. We have enhanced the limitations section of our paper to reflect on our choice of PubMed/MEDLINE. This addition acknowledges its focused nature and the potential exclusion of relevant studies from other sources. We trust this balanced approach, with your valuable input, enriches the depth and integrity of our work.

We selected PubMed/MEDLINE for its relevance in biomedical research, recognizing this choice narrows our review's scope. This might exclude studies from other databases, possibly limiting diverse insights. This limitation was considered to balance focus and comprehensiveness in our review.(page 23, lines 458-461, text highlighted)

  1. The problem statement describing GC as an issue should be more elaborate in the introduction section, discussing the demography of the sufferers of the disease and so on. 

Dear reviewer, thank you for your comment. According to your and reviewer 2 suggestion the introduction was changed and the following text was added:

The global prevalence of gastric cancer is increasing significantly, emphasizing the need for improved detection methods. Currently, gastric cancer ranks as the fifth most common malignant cancer and the fourth leading cause of cancer-related mortality worldwide. Despite a decline in incidence rates, the global burden of this malignancy is projected to increase by 62% by 2040 (2) . This escalation is further highlighted by predictions from the International Agency for Research on Cancer (IARC), which forecast an increase to about 1.8 million new cases and approximately 1.3 million deaths by 2040, representing increases of about 63% and 66%, respectively, compared with 2020.(3) ( page 5 lines 70-77, text highlighted)

  1. The review has concentrated on the DL techniques used in the classification, detection, and segmentation of GC. However, the authors could have included the optimal process or general framework used for the individual tasks and do a comparison of the processes. 

Dear reviewer, we appreciate your observation. Indeed, you are correct that the scope of AI is extensive and diverse. However, given that our review is titled 'Deep Learning and Gastric Cancer,' our emphasis was specifically on deep learning techniques. Considering the vastness of this subject, it's clear that additional reviews focusing on various other aspects are both necessary and valuable.

  1. Comparison of the data types should be done as well including the advantages and disadvantages and this can be linked to the algorithm used for each of the data types. for example, how do DL techniques used for CT scans differ from mammograms? How do they compare in terms of the statistical metrics used for evaluation?

Dear reviewer, our study focused on AI-assisted endoscopy, and therefore CT scans as well as mammograms are not within our scope. However, table 1 summarizes all AI models, performances and clinical detection objectives and clinical implications. 

  1. Authors should also discuss the issue of explainability of the DL model as there are lots of comparisons to the traditional endoscopist in the review

Dear reviewer, thank you for this comment. A text discussion the challenge of explainabilty was added to the discussion

Another challenge is the the explainability of DL)models, that  plays a pivotal role in clinical acceptance and decision-making. These DL models, trained on vast datasets of endoscopic images, have demonstrated high proficiency in identifying early signs of gastric cancer. However, the challenge lies in deciphering how these models arrive at their conclusions. For gastroenterologists, understanding the rationale behind a model's prediction is crucial for diagnosis and treatment planning. Explainability in this context involves the model highlighting specific features in endoscopic images, such as subtle changes in tissue texture, color, or vascular patterns, that signify pre-malignant or malignant changes. Techniques like Gradient-weighted Class Activation Mapping (Grad-CAM) (62) can be employed to visualize these decisive image regions, providing clinicians with a visual explanation of the model's diagnostic process. This level of transparency is essential, as it not only builds trust in the AI system but also aids in the educational aspect, helping clinicians to identify and understand subtle, early signs of gastric cancer. Thus, enhancing the explainability of DL models in AI-assisted endoscopy is a crucial step towards integrating these advanced tools into routine clinical practice for more effective and early detection of gastric cancer. ( page 21-22, lines 429-442, text highlighted)

  1. Authors have suggested future work to consider approaches that use multi-modal data for GC analysis. There is already lots of work in this area and authors may want to include this in their review as well.

 Dear reviewer, thank you for your comment. According to your suggestion, we expanded in the discussion the text regarding possible future research directions and further exploration.

In AI-assisted endoscopy for gastric neoplasia, researchers could focus on developing algorithms for enhanced detection and classification of early-stage lesions, offering real-time procedural assistance to endoscopists, and automating pathology correlation to potentially reduce biopsy needs. Further exploration could involve patient-specific risk assessments incorporating personal and genetic data, developing AI-driven post-endoscopic surveillance recommendations, and creating training tools for endoscopists. Additionally, integrating endoscopic data with other imaging modalities like CT or MRI could provide a more comprehensive diagnostic approach to gastric neoplasia. ( page 22, lines 446-452, text highlighted)

Round 2

Reviewer 2 Report

Comments and Suggestions for Authors

All comments/issues raised have been addressed. 

Reviewer 3 Report

Comments and Suggestions for Authors

The authors have answered all the questions and provided further information where appropriate. 

Accept the work as is.